# Modelling freckles and spurious grain formation in directionally solidified superalloy castings
Haijie Zhang [1], Yunxing Zhao[2], Wei Xiong[2], Dexin Ma[2], Andreas Ludwig[1], Abdellah Kharicha[1] & Menghuai Wu [1] ✉

Segregation channels with misoriented spurious grains, known as freckles, are an unacceptable casting defect in superalloy turbine blades. A digital-twin method to predict segregation channels was proposed in our previous studies; however, the formation of spurious grains was ignored. Here, we extend the digital twin methodology by incorporating dendrite fragmentation, which is recognized as the predominant mechanism in the formation of spurious grains. The flow-induced fragmentation process has been refined to account for the timing of dendrite pinch-off. A three-phase mixed columnar-equiaxed solidification model was used to track the motion of the crystal fragments. Directional solidification experiments for superalloy casting were conducted in an industrial-scale Bridgman furnace, and the distribution of spurious grains in the freckles was metallographically analysed. Excellent simulation-experiment-agreement was achieved. Based on this study, the formation of spurious grains within the segregation channels is mainly caused by the flow-driven fragmentation mechanism. Experimentally measured freckles can be reproduced only if the timing of the dendrite pinch-off is considered.

Superalloy single-crystal turbine blades are crucial components of aircraft engines and large land-based steam turbines owing to their superior high-temperature mechanical properties. The primary aim of achieving a single-crystal structure through directional solidification (DS) is to eliminate grain boundaries that are at a high risk of crack nucleation[1]. Freckles, which are characterised by a trail of randomly misoriented equiaxed (spurious) grains, second-phase particles, and pores, are intolerable defects in such components[2].

The mechanism of freckle formation is well understood. Thermosolutal convection caused by liquid density inversion in the mushy zone accounts for the onset of segregation channels[3,4]. The interdendritic liquid is less dense than the bulk liquid owing to the segregation of the alloying elements. With an upward DS, the light interdendritic liquid tends to rise, forming plumes at the solidification front[5,6]. By competing with each other, only a few plumes survive and develop into stable segregation channels in the mushy zone. The surrounding interdendritic liquid is sucked into the channels. Some roots of the high-order side branches of dendrites are pinched by remelting[7–9], and crystal fragments are produced. Based on experimental observations[3,10], tiny fragments (equiaxed grains) are blown out of the mushy zone by the upwelling flow and finally remelted in the upper hot-

melt region; coarse fragments sink and grow into misoriented spurious grains; and the residual solute-enriched liquid solidifies as eutectics. Traces of the local enrichment of eutectics and embedded spurious grains are termed freckles[11].

Despite previous experimental findings, predicting and controlling the occurrence of freckles in engineering processes remains challenging because the influencing factors are complex[12,13]. For example, process conditions with a high cooling rate and high temperature gradient are beneficial for suppressing the formation of freckles[11,12]; casting with a contracted cross-sectional shape is more prone to freckles[13,14]; and the shadow side of a casting in the furnace, especially at the casting corner, is the most favourable position for the onset of freckles[13,15]. These empirical rules provide valuable insights into freckle formation but are not sufficient for making quantitative predictions. The Rayleigh number ($Ra$) was used to evaluate the freckling potential[4,16]. Recently, the DS process of superalloys was directly investigated using in-situ and real-time X-ray techniques[17]. Segregation channels were formed on the casting surface because of the accumulation of solute-enriched liquid. Dendrite fragmentation events were detected within these channels; however, the fragments were only observed several minutes after the initial appearance of the channels. Some of the as-formed fragments floated, and some sank. Finally, the fragments solidified into spurious grains

[1]Chair for Simulation and Modelling of Metallurgy Processes, Department of Metallurgy, University of Leoben, A-8700 Leoben, Austria. [2]Central South University, Powder Metallurgy Research Institute, 410083 Changsha, China. ✉e-mail: menghuai.wu@unileoben.ac.at

in the segregation channels. The *Ra* criterion fails to consider such complex flow-solidification interactions.

Numerical modelling exhibits its capability to quantify the formation of freckles. Some solidification models that directly solve dendrite morphology at the microstructure level offer compelling explanations for the initiation of freckles[18–20]. However, the extremely high computational cost limits these solidification models to a small portion of the casting, for which the overall flow in the casting and the thermal field in the furnace have to be simplified or ignored. For the time being, such models are unsuitable for simulating engineering casting. Recent studies suggest that the volume-average solidification model is the most promising approach for engineering applications[21]. The solidification physics that occur on the microscopic scale, including the solute partition at the solid-liquid interface, release of latent heat, and momentum exchange due to phase change, are volume-averaged and coupled with macroscopic transport equations. The volume-average solidification model has been successfully used to study the onset and further development of segregation channels during the DS of superalloys[22–24]. Combining the volume-average solidification model in DS casting with a global temperature field calculation in an industrial furnace, the current authors previously proposed a digital twin approach to predict the segregation channels in an engineering-scale turbine blade[25]. Although real-time interaction with the physical process remains unattainable, it is possible to create a digital replica of the physical process. Despite this promising progress, the formation of spurious grains has not been considered. The calculated segregation channels were not identical to freckles. Gu et al.[26] simulated the motion and remelting of fragments during the DS of superalloy casting. Although the fragmentation process was not solved directly, their work highlighted the importance of considering fragments in freckle prediction by assuming an arbitrary initial fragment size and location.

Dendrite fragmentation, that is, detachment of the side arms from the dendrite main stem, is believed to be the main mechanism for the origin of spurious grains in freckles[3,11]. Investigations into dendrite fragmentation date back to the 1960s[7]. Although mechanical fracture was initially regarded as a mechanism for dendrite fragmentation[27], dendrite fragmentation is now widely accepted to have originated from the remelting of the roots of side branches[7–9,28,29]. One reason for remelting is melt convection, which causes local fluctuations in the thermal and/or solute fields in the interdendritic region[7,30]. Another reason for remelting is capillary-driven coarsening[31]. Diffusion of the solute in the liquid leads to dendrite growth in regions of lower curvature. This occurs at the expense of regions with higher curvature[8]. Therefore, two fragmentation submodels have been proposed: the flow-driven fragmentation model[32] and the capillary-driven fragmentation model[33]. Both submodels have been used to analyse the solidification of different casting processes[34,35]. The fragments produced via the flow-driven fragmentation mechanism mostly contributed to the final as-cast structure under strong flow conditions. However, the timing of the dendrite

pinch-off by the flow-driven fragmentation mechanism was not considered, which may cause errors in estimating the predicted fragments.

In the current study, the previously proposed digital twin method[25] is extended by implementing dendrite fragmentation to investigate the formation of spurious grains and freckles during the DS of superalloy casting. Two crystal fragmentation mechanisms are considered: flow-driven and capillary-driven. A $t_r - \tau$ approach is suggested to treat the timing of the dendrite pinch-off. A three-phase mixed columnar-equiaxed solidification model[21,36] is used to model the solidification process and track the fragment motion. To validate the model, DS experiments for superalloy casting were conducted. A cluster of superalloy castings was assembled around the axis of the furnace; however, in this work, only one-twelfth of the furnace, including a single superalloy casting with a cross-sectional variation, was simulated. Based on this study, the formation of spurious grains within the segregation channels is mainly caused by the fragments produced by the flow-driven fragmentation mechanism. Experimentally measured freckles can be reproduced only if the timing of the dendrite pinch-off is considered. The shadow effect of the Bridgman furnace provided the most favourable locations for the formation of freckles. The geometrical effect on freckle formation was also confirmed; the expansion of the cross-section suppressed the development of freckles, whereas the contraction of the cross-section promoted freckle formation.

## Results
### Experimental analysis of the casting
After the solidification process, the as-solidified casting was then removed from the ceramic mould. The entire casting was blast-cleaned, and macroetching was performed to examine the distribution of freckles on the casting surface. The shadow side of the casting is prone to freckle formation, while the other three sides remain freckle-free. The surface of the shadow side of the as-solidified casting is shown in Fig. 1a. A schematic of the freckle distribution on the casting surface is presented in Fig. 1b. Freckles in both the small and large cross-sectional segments were detected only on the shadow side. The top part of the casting was more prone to freckle formation than the lower part. Additionally, the freckles started from the bottom of each contracted segment and were formed after an incubation period in each expanded segment. Metallographic analysis was conducted to show more details about the freckles. The casting vertical surfaces, marked by A1 to A3 in Fig. 1a, was polished and re-etched. The as-cast structures depicted in Fig. 1c–e were obtained using the light microscopy. The primary columnar dendrites (in yellow and brown), eutectics between dendrites (in white), and freckles (in dark brown and black) can be distinguished. The dark-coloured freckles originated from the segregation channels, which are marked by the red dashed lines in Fig. 1e. The Electron Backscatter Diffraction (EBSD) detection was performed on the region marked in blue in Fig. 1e. The detected results are shown in Fig. 1f. Numerous small spurious

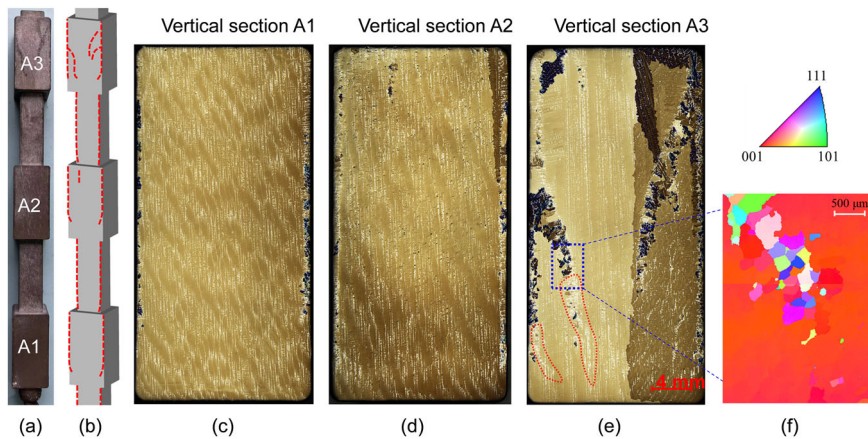

**Fig. 1 | Experimental analysis of the as-cast structure. a** Distribution of freckles on the shadow side of the casting. **b** A schematic of the overall distribution of freckles on the casting surface. **c**–**e** Microstructure on the vertical sections A1 to A3 marked in Fig. 1a. **f** Electron Backscatter Diffraction detection.

**Fig. 2 | Temperature results. a** Temperature distribution in the furnace. **b** Temperature distribution on the casting surface. **c** Cooling histories at two specific points A (heading side) and B (shadow side), as marked in (**b**); the temperature difference between the two points ($T_A$-$T_B$) is also plotted.

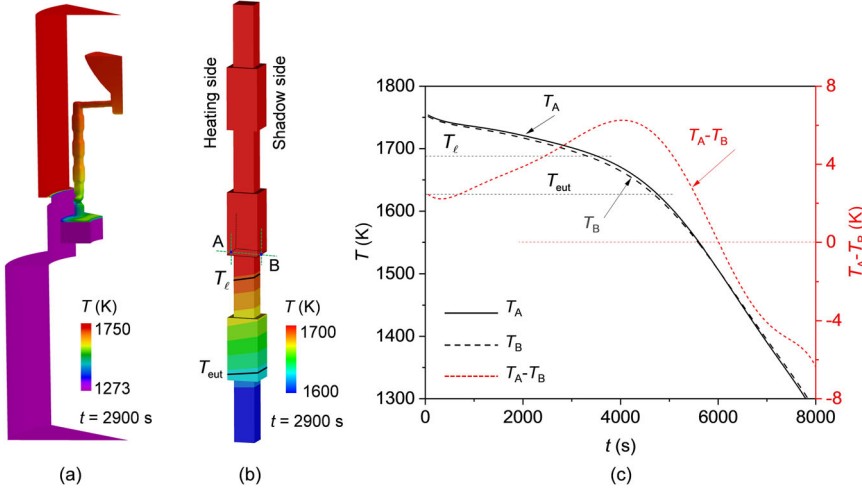

**Fig. 3 | Solidification sequence at 2200 s. a** Contour of solid fraction ($f_s$) on the lower half of the casting overlaid with the solidification front ($f_s = 0.01$) and the eutectic isotherm ($T = T_{eut}$). The liquid flow on the central vertical section is also shown, with flow magnitude ($|\mathbf{u}_\ell|$) indicated by the colour of the arrows. The yellow dashed lines indicate the flow directions. **b** Contour of liquid concentration ($c_\ell$) on back (shadow side) and right walls of the casting. **c** Contour of liquid density ($\rho_\ell$) on the back and right walls of the casting. Isotherms are overlaid. **d** Yellow and red isosurfaces of macrosegregation index ($c_{mix}^{index} = 2.6\%$, with $c_{mix}^{index} = 100 \cdot (c_{mix} - c_0)/c_0$) showing the shape of plumes and segregation channels, respectively.

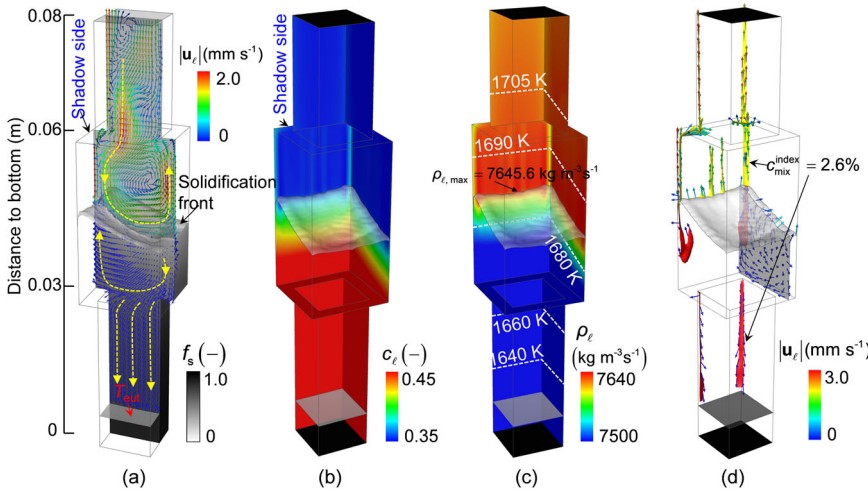

grains are dispersed within the freckles, whereas the segregation channels are free of spurious grains but exhibit an accumulation of eutectics.

## Thermal calculation with ProCAST

The calculated temperature in the furnace and on the casting surface of the superalloy is shown in Fig. 2a, b, respectively. Because of the shadow effect of the Bridgman furnace, the heating side of the casting facing the induction wall was warmer than that facing the shadow side. The cooling histories of the superalloys at two specific points A and B, as indicated in Fig. 2b, along with their temperature difference, are illustrated in Fig. 2c. The alloy was gradually cooled from 1750 K. A temperature difference, decreasing from 6 K to −6 K, developed during the withdrawal process. Such a temperature difference led to an inclined solidification front. The solidification front on the shadow side was 6.4 mm more advanced in comparison to the heating side.

## Flow and solidification calculation using the volume-average solidification model

The solidification sequence at 2200 s is shown in Fig. 3. Owing to the shadow effect in the Bridgman furnace and the inclined solidification front, as depicted in Fig. 3a, a large circular flow pattern was developed in the bulk liquid above the solidification front. The bulk liquid flowed downward on the shadow side and upward on the heating side. The flow patterns in the mushy zones also differed. In the upper part of the mushy zone, the interdendritic liquid flowed downward on the heating side and upward on the

shadow side. In the lower part of the mushy zone (in the small cross-sectional region of the casting), the interdendritic liquid was sucked downward to feed the solidification shrinkage. Because of solute partitioning during solidification, the solute was continuously rejected into the interdendritic liquid, leading to an increase in solute concentration ($c_\ell$) with the depth of the mushy zone (Fig. 3b). Noticeable light blue strips were observed above the solidification front, which indicate a high $c_\ell$. The change in the liquid density ($\rho_\ell$) is depicted in Fig. 3c. Both solutal and thermal expansions determine the value of $\rho_\ell$, which decreased in the bulk region with casting height owing to thermal expansion and in the mushy zone with depth owing to solutal expansion. The liquid immediately above the solidification front exhibited the highest value of $\rho_\ell$. This density inversion led to hydrodynamic instability. The less dense liquid within the mushy zone tended to flow upward. As depicted in Fig. 3d, above the solidification front, two prominent plumes (yellow parts) were developed at the shaded corners of the casting, accompanied by the formation of several smaller, thinner plumes between them. Below the solidification front, two segregation channels (red parts) formed beneath the two prominent plumes. The liquid flowed upward within the plumes and channels. In return, the liquid flow further updated the $c_\ell$ and $\rho_\ell$ profiles by sucking the solute-enriched liquid into the channels from their surrounding interdendritic area. The solute-enriched liquid was then transported upward in the channels and plumes.

The evolution of the plumes is shown in Fig. 4. At 1500 s, plumes developed around the solidification front, with strong plumes at the four corners and weak plumes between them (Fig. 4a). The liquid within the

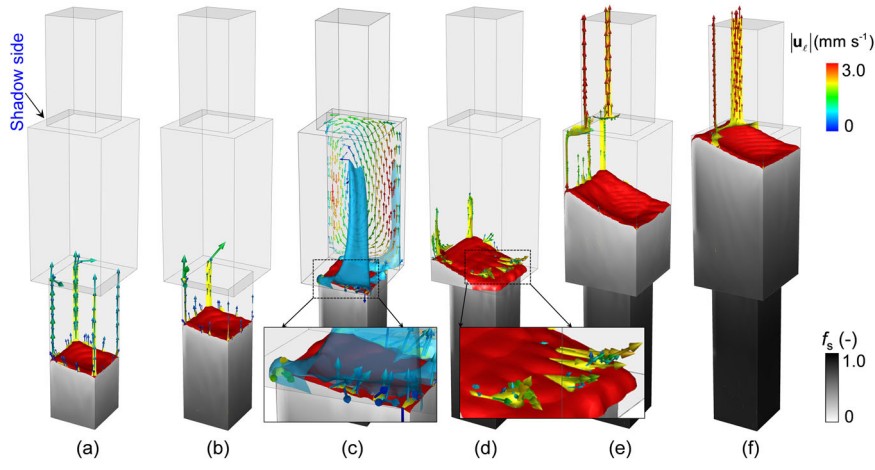

**Fig. 4 | Evolution of plumes during the solidification process.** The grey scale denotes $f_s$, the red isosurfaces ($f_s = 0.01$) indicate the solidification front, and the yellow isosurfaces $c_{mix}^{index} = 2.6\%$ indicate the as-developed plumes. Blue isosurfaces of $c_{mix}^{index} = 2.5\%$ are overlaid in (**c**) to demonstrate the solute trace. The coloured vectors denote the liquid flow. **a** Results at 1500 s. **b** Results at 1660 s. **c** Results at 1930 s. **d** Results at 2000 s. **e** Results at 2220 s. **f** Results at 2460 s.

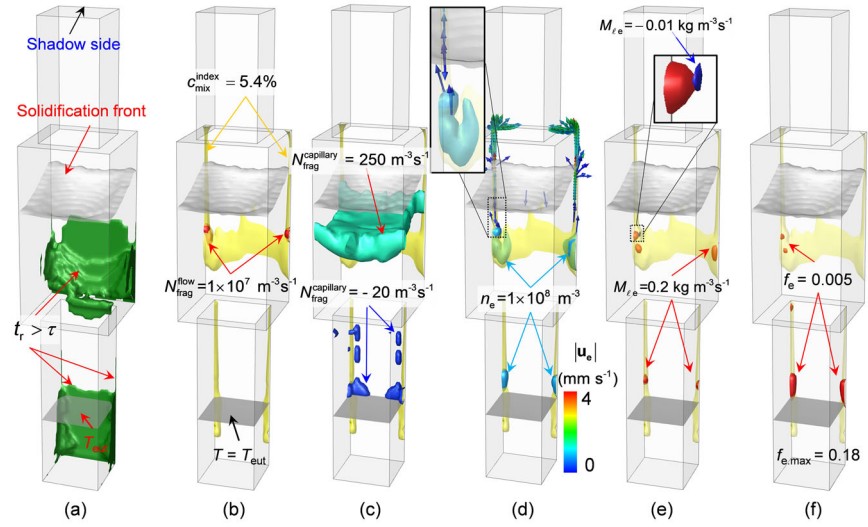

**Fig. 5 | Formation mechanism of spurious grains at 2320 s.** The solidification front is denoted by the grey isosurface of $f_s = 0.01$, and the bottom of the mushy zone is denoted by the isotherm of $T = T_{eut}$. The segregation channels are indicated by yellow isosurfaces of $c_{mix}^{index} = 5.4\%$. **a** Potential flow-driven fragmentation region determined by the timing of dendrite pinch-off. Green isosurfaces of $H_x = 1$. **b** Fragment production rate by the flow-driven fragmentation mechanism. Red isosurfaces of $N_{frag}^{flow} = 1 \times 10^7 \text{m}^{-3}\text{s}^{-1}$. **c** Fragment production rate determined by the capillary-driven fragmentation mechanism. Light blue isosurfaces of $N_{frag}^{capillary} = 250 \text{ m}^{-3}\text{s}^{-1}$ indicate production of fragments, and dark blue isosurfaces of $N_{frag}^{capillary} = -20 \text{ m}^{-3}\text{s}^{-1}$ indicate destruction of fragments by dendrite coarsening. **d** Number density of spurious grains. Light blue isosurfaces of $n_e = 1 \times 10^8 \text{m}^{-3}$. The vectors show the movement of grains. **e** Solidification/remelting rate of spurious grains. Solidification is indicated by red isosurfaces of $M_{\ell e} = 0.2 \text{ kg} \cdot \text{m}^{-3}\text{s}^{-1}$, and remelting is indicated by the dark blue isosurface of $M_{\ell e} = -0.01 \text{ kg} \cdot \text{m}^{-3}\text{s}^{-1}$. **f** Distribution of the volume fraction of spurious grains $f_e$ with red isosurfaces of $f_e = 0.005$.

plumes flowed upward. As the solidification proceeded ($t = 1660$ s), the two plumes at the corners of the shadow side persisted and became thicker, whereas those at the corners of the heating side became invisible (Fig. 4b). When the solidification front approached the cross-section expansion at $t = 1930$ s, the plumes attempted to extend into the large section of the casting, but they were blown away by the strong thermal convection in the bulk. This can be identified from the solute trace represented by the blue isosurfaces ($c_{mix}^{index} = 2.5\%$) in Fig. 4c. Some short plumes could still be observed around the solidification front, and they were all inclined to the right side. At 2000 s (Fig. 4d), two new plumes appeared at the shadow corners. It should be noted that the two plumes were not simple extensions of the old ones, originating from the small cross-sectional region. The old ones were killed by strong thermal convection when they underwent cross-sectional expansion. At 2220 s, Fig. 4e, the two new plumes on the shadow side continued to grow, and the upward flow intensity of the plumes was almost twice as strong as that observed for the plumes present in the part of the casting with the smaller cross-section in Fig. 4b. When the solidification front approached the cross-sectional contraction area at 2460 s (Fig. 4f), the two plumes survived the cross-sectional contraction, bent, and continued to grow into the smaller cross-section casting region.

## Fragmentation mechanism to form spurious grains

As described in the method section, crystal fragmentation by the flow-driven mechanism occurs when the following two conditions are fulfilled: (1) the flow-driven remelting condition, $(\mathbf{u}_\ell - \mathbf{u}_c) \cdot \nabla c_\ell < 0$, and (2) the long enough accumulated remelting time $t_r > \tau$ with $t_r = \sum_{i=1}^{\infty} \Delta t_i |_{(u_\ell - u_c)\nabla c_\ell < 0}$. The modelling result for the formation of spurious grains at 2320 s is analysed in Fig. 5. The region that fulfils the second condition ($t_r > \tau$) is shown in Fig. 5a. By combining with the first condition, the locations for the flow-driven fragmentation can be predicted. As depicted in Fig. 5b, the fragments were only produced around the segregation channels (shown by the yellow isosurfaces), with $N_{frag,\ max}^{flow} = 2.1 \times 10^8 \text{ m}^{-3}\text{s}^{-1}$. Differently, the capillary-driven fragmentation mechanism mainly operated in the central area of the casting (Fig. 5c), and an ignorable production rate was predicted; that is, $N_{frag,\ max}^{capillary} = 430 \text{ m}^{-3}\text{s}^{-1}$. The negative $N_{frag}^{capillary}$ at the lower half of the mushy zone indicates the destruction of spurious grain through coarsening[8,31]. The number density of spurious grains ($n_e$) produced by both fragmentation mechanisms is shown in Fig. 5d. Most of the spurious grains were located near the segregation channels. The maximal $n_e$ was $7.6 \times 10^9 \text{ m}^{-3}$. According to the inset in Fig. 5d, some spurious grains

**Fig. 6 | Comparison between the simulation and experimental results for freckles and spurious grains. a** Volume fraction of columnar phase $f_c$. **b** Volume fraction of eutectic phase $f_{eut}$. **c** Volume fraction of equiaxed phase $f_e$. **d** Macrosegregation index $c_{mix}^{index}$. **e** Distribution of segregation channel (the green isosurfaces, $c_{mix}^{index} = 5.4\%$) and spurious grains (the red isosurfaces, $f_e = 0.01$). **f** Distribution of freckles on the casting surface after macro-etching. **g** Microstructure on the horizontal cross-section (A-A) along the red dashed line in (**f**). **h** The calculated $c_{mix}^{index}$ distribution on the same A-A section.

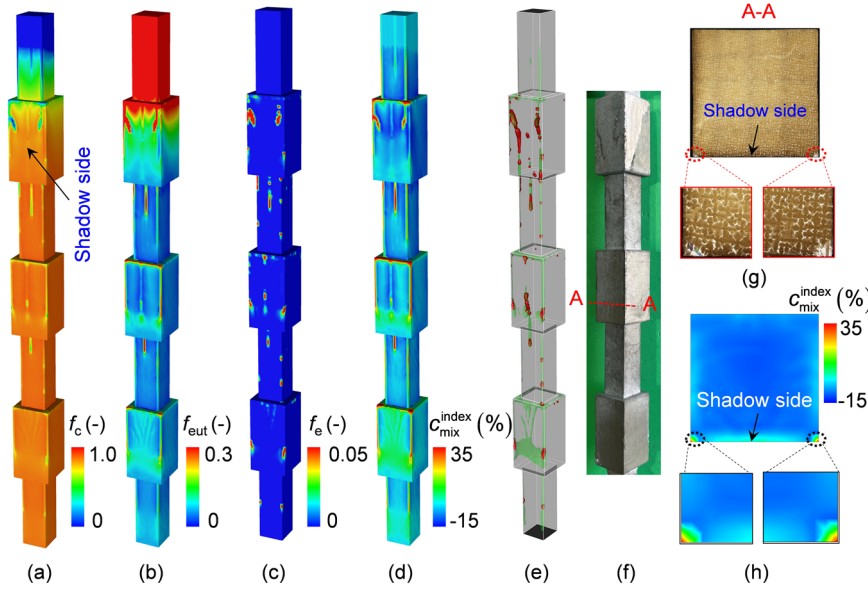

moved upward along with the upwelling liquid in the segregation channels. Grain settling was also predicted, but only a very small number of grains appeared to sink. Because most of the spurious grains were produced deep in the mushy zone, they were immediately obstructed by the local columnar dendritic structure after their origin. The as-formed spurious grains underwent growth or remelting. The mass transfer rate between the liquid and the spurious grains, $M_{\ell e}$, is shown in Fig. 5e. The red isosurfaces defined by $M_{\ell e} = 0.2\,\mathrm{kg \cdot m^{-3} s^{-1}}$ indicate the solidification of the grains. Because the solidification rate of spurious grains is proportional to $n_e$[36], a large $M_{\ell e}$ was observed in the regions where $n_e$ was large. The blue isosurface in the inset in Fig. 5e represents the remelting process, i.e., $M_{\ell e} = -0.01\,\mathrm{kg \cdot m^{-3} s^{-1}}$. Compared to the solidification process, remelting occurred only occasionally throughout the whole casting process. The volume fraction of remaining spurious grains (Fig. 5f) reached 0.18 in the segregation channels at the moment but was negligible (<0.001) in the rest of the casting. After the casting has fully solidified, the solute-enriched liquid in the segregation channels solidifies as eutectics, with spurious grains embedded within it, a phenomenon referred to as freckles.

### Comparison with experiments

Figure 6 shows a comparison between the simulated as-cast structure and freckle distribution and the experimental results. In the freckle-free area, $f_c \approx 0.89$ (volume fraction of columnar phase) and $f_{eut} \approx 0.11$ (volume fraction of eutectics). Inside the freckles, the maximum $f_e$ value reached 0.5 at some locations, and the rest of the structure was composed of eutectics and columnar dendrites. As illustrated in Fig. 6d, positive macrosegregation was observed in the freckles. To show the spurious grains inside the segregation channels, the outer surface of the casting is shown to be transparent in Fig. 6e. The red isosurfaces show regions with spurious grains, and the green isosurfaces show traces of segregation channels. Two relatively strong segregation channels were observed along the edges on the shadow side of the casting, and some short/weak segregation channels were observed between them. Comparing the simulation with the experimental results (Fig. 6f and Fig. 1), all spurious grains were dispersed in the segregation channels, into which eutectics had also accumulated. As the casting is withdrawn, the downward movement of the water-cooled copper plate reduces the isolation between the furnace's hot and cold zones. This decreases the temperature gradient at the solidification front, creating more favourable conditions for the formation and development of freckles. Consequently, the upper part of the casting was more susceptible to freckles than the lower part. The spurious grains detected in the middle-width area

on the shadow side in Fig. 1d were also successfully predicted. Additionally, both the experiment and simulation showed that freckles were formed immediately when the cross-section contracted, whereas an incubation distance of approximately 5 mm was needed to form freckles when the cross-section expanded. The microstructure of the horizontal cross-section cut along the red dashed line in Fig. 6f is compared in Fig. 6g, h. An excellent agreement between the simulation and experimental results was achieved.

### Discussion

According to the engineering definition, freckles are different from channel segregation. A segregation channel is called a freckle only if it captures spurious grains[11]. A segregation channel without spurious grains can at most be called a 'quasi-freckle'. In the past, most modelling efforts were focused on channel segregation[22,23,37,38], i.e., the formation of spurious grains was ignored. Gu et al.[26] investigated the channel flow and its effect on the motion and fate (survival or remelting) of a fragment whose origin was arbitrarily set. Large fragments, or fragments initially positioned deep in the segregation channel, sank and survived in the channel, whereas small fragments were advected out of the open segregation channels and completely remelted in the hot bulk liquid. This preliminary study raises a new question: When the effect of channel flow on the fate of spurious grains is known, do the spurious grains inside a channel influence the fate of a segregation channel?

To answer this question, a new simulation was performed in which dendrite fragmentation was ignored. It was then compared with the previous simulation with dendrite fragmentation (Fig. 7). Quantitatively, the shapes and numbers of channels were not identical for both simulation cases. The global segregation intensity was also evaluated. The global segregation index over the entire casting volume (GMI = $\oint_{Vol} |c_{mix}^{index}| dV / \oint_{Vol} dV$) was 3.29% for the case in which fragmentation was ignored, which is similar to the value of 3.32% for the case with fragmentation. Considering spurious grains has a very limited effect on the simple-shaped casting with cross-sectional changes, but it has a significant effect on the industrial-scale turbine blade casting with complex geometries and thin-wall structures. More discussion is provided below.

As solidification can cause a volume shrinkage of 8.26%, the feeding flow effect is not negligible for the current superalloy. Deep in the mushy zone (Fig. 3a), only a parallel downward flow was observed. This shows that the feeding flow dominated because thermosolutal convection was damped by the dendritic structure of the mushy zone. Such a feeding flow in the deep mushy zone does not induce any new channels or freckles, but it can influence the further growth or reduction of existing channels/freckles. Near

the solidification front, at the critical location for the onset of plumes or channels, thermosolutal convection dominated, and the contribution of the feeding flow was imperceptible. However, the feeding flow effect can be intensified or reduced at certain positions, depending on the casting geometry. For example, in castings in which the cross-section changes, the feeding flow effect increases or decreases in linear proportion to the ratio of the cross-sectional areas[39]. To study this feeding flow effect in combination with the geometrical cross-sectional change, an extra case was simulated in

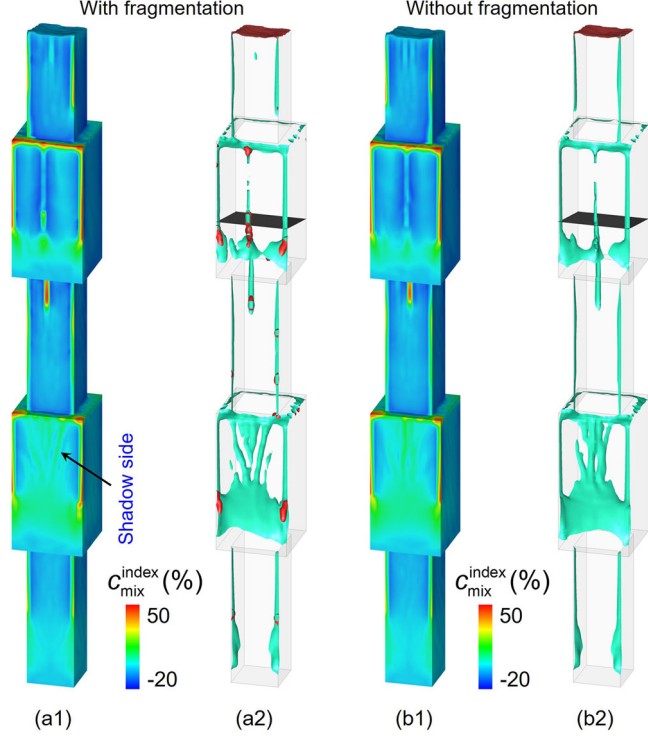

**Fig. 7 | Comparison of freckles and channels calculated by considering and neglecting dendrite fragmentation. a1**, **a2** Results when fragmentation is considered. **b1**, **b2** Results when fragmentation is ignored. **a1**, **b1** Contour of macrosegregation index $c_{\mathrm{mix}}^{\mathrm{index}}$ on the surface of the casting. **a2**, **b2** Freckles and channels. The predicted freckles indicated by the segregation channels (light-blue isosurfaces of $c_{\mathrm{mix}}^{\mathrm{index}} = 5\%$) and a chain of spurious grains (red isosurfaces of $f_e = 0.01$).

which solidification shrinkage was ignored. The results were compared with those of the previous case, in which solidification shrinkage was considered.

The flow streamlines at 2600 s in both cases are compared in Fig. 8a, b. A clear difference is observed in the lower half of the mushy zone. As shown in Fig. 8a, the feeding flow overwhelmed the natural convection at a height of approximately 50 mm; hence, a downward flow was predicted over the entire section of the casting. In contrast, when solidification shrinkage was ignored, as shown in Fig. 8b, the liquid flowed upward along the channels throughout the mushy zone. The less-dense liquid in the lower part of the mushy zone flowed upward. Therefore, at the height of 57 mm, the upward flow in the segregation channels increased from 0.09 mm/s for the case in which solidification shrinkage was considered to 1.0 mm/s for the case in which solidification shrinkage was ignored. A significant difference in the number of segregation channels/freckles in the as-solidified casting ($t = 5000$ s) was predicted between the two simulation cases, as shown in Fig. 8c, d. When the feeding flow is ignored, the upward flow in the segregation channels is significantly overestimated. When the solidification front approaches cross-section expansion, the overestimated upward flow enables the plumes to withstand the impact of thermal convection; hence, the two channels initiated in the small section can grow continuously into the large section part (Fig. 8d). The overestimated liquid flow also provides a more favourable condition for dendrite fragmentation; thus, more spurious grains are formed in the segregation channels when solidification shrinkage is ignored.

The geometrical effect on freckle formation in superalloy castings has been experimentally demonstrated[13–15], but the complex interplay between the thermo-solutal convection and feeding flow was not explained. The current modelling tool provides a quantitative explanation to the geometrical effect.

To the best of our knowledge, this is the first study to consider the timing of dendrite pinch-off for a flow-driven fragmentation mechanism in a numerical model. Utilising the newly extended fragmentation model, a satisfactory agreement between the simulation and experimental results was achieved for superalloy casting. Other fragmentation models[28,32] have been used for current casting; however, the calculated spurious grains and freckles were unsatisfactory.

As shown in Eq. (10)[40], the characteristic time ($\tau$) for pinch-off is material- and dendrite-shape-dependent. Some data are available from the literature to calculate $\tau$, i.e., $f(0) = 0.6$, $\lambda_0 = 2.27$ nm, and $D_\ell = 3.6 \times 10^{-9}$ m$^2$ s$^{-1}$[22,40]. However, the initial root diameter of the side arms ($d_{\mathrm{pr},t_0}$) is difficult to estimate. Theoretically, if the as-solidified dendrites characterise a thick root diameter, the remelting time to pinch off the roots is long, and, therefore, both the production rate of the fragments and the volume fraction of the formed spurious grains are low. In this study, test simulations were

**Fig. 8 | Study of the feeding flow and geometrical effects on the formation of freckles.** Two simulations are compared: (**a**, **c**) considering solidification shrinkage; (**b**, **d**) ignoring solidification shrinkage. **a**, **b** Flow patterns at 2600 s. **c**, **d** Predicted freckles in the as-solidified casting at 5000 s. The blue isosurfaces indicate the solidification front ($f_s = 0.01$), and the black isosurfaces indicate the eutectic isotherm ($T = T_{\mathrm{eut}}$). The red streamlines show the flow pattern. The segregation channels are indicated by the green transparent isosurfaces ($c_{\mathrm{mix}}^{\mathrm{index}} = 5.4\%$), and the red isosurfaces ($f_e = 1.0\%$) show the distribution of spurious grains.

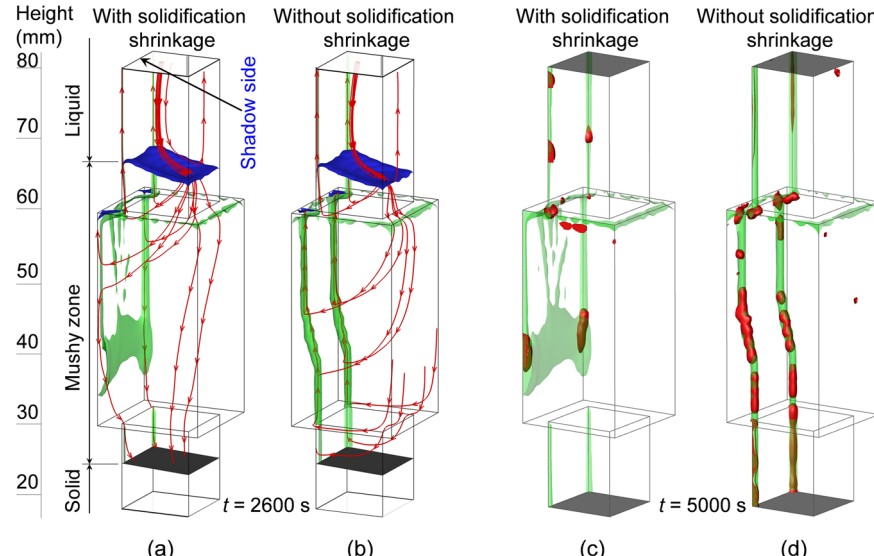

**Fig. 9 | Application of the newly extended model to an industrial single-crystal turbine blade casting.** **a** Calculated segregation channels on the convex side and a horizontal section of the casting. **b** The calculated $f_e$ on the convex side and a horizontal section of the casting. **c** Trajectories of the segregation channels indicated by green iso-surfaces ($c_{mix}^{index} = 10\%$) and the distribution of spurious grains indicated by red iso-surfaces ($f_e = 0.05$). **d** Experimental measurement of the turbine blade casting.

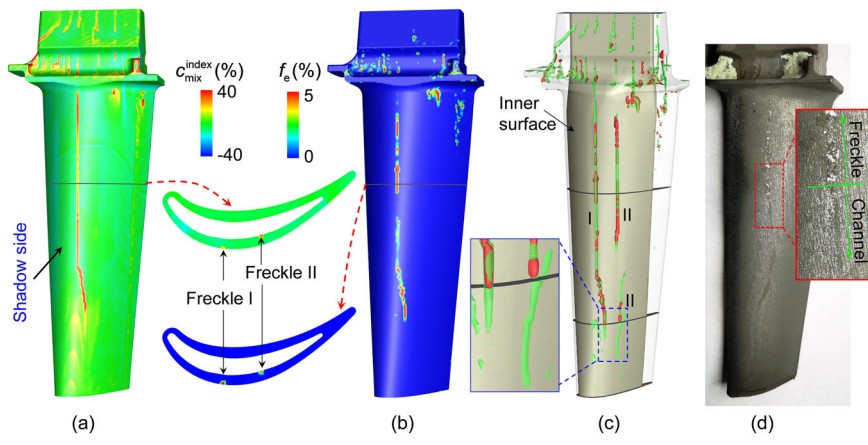

performed by changing $d_{pr,t_0}$ to fit the simulation results to the experimental measurements. When $d_{pr,t_0} = 12.8\,\mu m$, the simulation results matched the experimental results. The value $d_{pr,t_0} = 12.8\,\mu m$ agreed well with the results of a quenching solidification experiment[41] (CMSX-4, cooling rate of 3 mm/min, temperature gradient of 5 K/mm), where $d_{pr,t_0}$ was found to be in the range of 8.8–16.5 μm. Note that $\tau$ must be adapted if the current fragmentation model is applied to other alloys.

During the solidification process, the side arms of columnar dendrites may undergo solidification and remelting alternatively, depending on the sign of $(\mathbf{u}_\ell - \mathbf{u}_c) \cdot \nabla c_\ell$. When $(\mathbf{u}_\ell - \mathbf{u}_c) \cdot \nabla c_\ell < 0$, $d_{pr}$ is reduced via remelting. Conversely, during the time steps marked with red squares, i.e., when $(\mathbf{u}_\ell - \mathbf{u}_c) \cdot \nabla c_\ell > 0$, $d_{pr}$ may increase via solidification. The net local remelting time should be a balance of the solidification and remelting times. In the currently studied superalloy, after the development of channel segregation, the flow within the channel nearly reached a quasi-steady state, where it sustained the remelting condition. However, if the current approach is to be applied to other solidification techniques, careful consideration of the net local remelting time is essential.

In our previous study[25], the directional solidification of an industrial single-crystal turbine blade was calculated using a two-phase solidification model in which the formation of spurious grains was not involved. That turbine blade casting was produced using the same superalloy, and the processing parameters were identical to those for the currently studied sample. Therefore, the newly extended three-phase solidification model and verified fragmentation parameters were applied directly to simulate the solidification of the turbine blade. The simulation results for the case when the casting had fully solidified are depicted in Fig. 9. The macrosegregation index was very similar to the results obtained from the two-phase solidification model, which was validated by experimental measurements[25]. Figure 9b shows the calculated distribution of spurious grains. Spurious grains were generated in the segregation channels. A very small number of spurious grains were produced via the capillary-driven fragmentation mechanism, as demonstrated in Fig. 5c, but their volume fraction was negligible. In Fig. 9c, the trajectories of the segregation channels are superimposed on the distribution of spurious grains, and the blade is rendered transparent. Spurious grains were predicted in all channels and were distributed discretely within the channels. An experimental inspection after the macroetching is shown in Fig. 9d. A long freckle developed on the convex side of the cast. Spurious grains are visible in the top part, whereas only a segregation channel without visible spurious grains is observed in the lower part. The same phenomenon can be seen in Fig. 1e, where the segregation channels below the freckles are delineated by red dashed lines. The phenomenon that a freckle is initiated from an as-developed segregation channel was also confirmed numerically. As shown in the magnified view in Fig. 9c, spurious grains were generated only after the segregation channels were well developed.

The newly extended multiphase volume-average solidification model was successfully applied to an industrial turbine blade, and satisfactory agreement between the simulation and experimental results was achieved.

Compared to our previous research[25], it was found that considering spurious grains has a significant impact on the prediction of channel segregation. In earlier two-phase simulations that ignored spurious grains, a segregation channel was predicted on the concave side of the blade casting, but this channel was not observed experimentally. The current three-phase model, which accounts for spurious grains, resolves this discrepancy between experimental results and simulation predictions. Another major difference lies in the shape and distribution of the segregation channels on the convex surface. In Fig. 9a, the fish-tail-shaped region (in yellow colour) at the bottom of the casting indicates the competition and merging of plume flows during solidification[2,11] before developing into a stable segregation channel. This phenomenon was not predicted by the two-phase model. The pronounced effect of considering the spurious grains on the prediction of channel segregation in blade castings may be attributed to the fact that, in thin-walled castings, even a minor influence of spurious grains on fluid flow can significantly impact the initiation and further development of freckles. Essentially, neglecting the formation of spurious grains only allows for the prediction of channel segregation. It is only by considering the formation, movement, and solidification of spurious grains that freckles can be truly predicted.

However, a quantitative mismatch between the simulation and measured results remained. As shown in Fig. 9, the calculated freckles are longer than the experimental measurements. There are several possible reasons for this discrepancy. First, the characteristic time ($\tau$) for pinch-off, which is a function of the dendrite morphology and diameter of the side arm roots, should be a variable. However, this correlation has not yet been confirmed. Second, simplifying a multicomponent alloy as a binary alloy may introduce inaccuracies[42]. Third, most material properties are temperature- and solute-dependent. Unfortunately, most of these data are not available in the literature. As the simulation and experimental results were in good agreement, the mesh size used for the simulation should be acceptable. Using this mesh size, one simulation on a high-performance cluster (2.6 GHz and 28 cores) required approximately one month to complete. Simulations with a smaller mesh size are not feasible with the current computation capabilities.

## Conclusions

The digital twin method proposed by the current authors was extended by implementing dendrite fragmentation to investigate the formation of spurious grains and freckles during DS of superalloy casting. Two fragmentation mechanisms were considered: flow-driven and capillary-driven. A $t_r - \tau$ criterion was proposed to treat the timing of dendrite pinch-off by the flow-driven fragmentation mechanism. A three-phase mixed columnar-equiaxed solidification model, considering the interaction between the grain motion and liquid flow within the segregation channels, was used to model the solidification process. To validate the model, a DS experiment for superalloy casting with a varying cross-section was conducted in an industrial-scale Bridgman furnace. A satisfactory agreement between

## Table 1 | Equations for grain transport, grain-destruction and fragmentation

| Equations | | Symbols |
|---|---|---|
| **1. Conservation of the number density of equiaxed grains** | | |
| $\frac{\partial}{\partial t} n_e + \nabla \cdot (\mathbf{u}_e n_e) = N_{frag}^{capillary} + N_{frag}^{flow} + N_{des}$ | (1) | $n_e$: number density of equiaxed grains. $u_e$: velocity of equiaxed grains. $t$: physical time. $N_{frag}^{capillary}$: fragmentation rate by capillary-driven mechanism. $N_{frag}^{flow}$: fragmentation rate by flow-driven mechanism. $N_{des}$: grain destruction rate. |
| **2. Grain destruction via remelting** | | |
| $\frac{d(n_e)}{d(d_e)} = \frac{n_e}{\sqrt{2\pi}\sigma d_e} e^{-\frac{1}{2}\left(\frac{\ln(d_e)-\ln(\hat{d}_e)}{\sigma}\right)^2}$ | (2) | $\sigma$: geometric standard deviation of the lognormal distribution. $\hat{d}_e$: geometric mean of the grain diameter. $d_e$: grain diameter. $v_{R_e}$: grain growth/remelting speed. $d_{e,critical}$: critical value of the grain size, below which the grain is destructed (eliminated). $D_\ell$: diffusion coefficient in liquid. $l_\ell$: diffusion length in liquid. $k$: partition coefficient. |
| $\frac{d(d_e)}{d(t)} = v_{Re} \cdot \frac{D_\ell}{l_\ell} \cdot \frac{(c_\ell^* - c_\ell)}{(1-k)c_\ell^*}$ | (3) | |
| $N_{des} = v_{Re} \cdot \frac{d(n_e)}{d(d_e)}\Big|_{d_e = d_{e,critical}}$ | (4) | |
| **3. Capillary-driven fragmentation** | | |
| $N_{frag}^{capillary} = \frac{d(a \cdot S_V^3)}{dt}$ | (5) | $a, \bar{r}, S_{S0}, K_0$: alloy-dependent constants. $t_f$: local solidification time. $M_{frag}^{capillary}$: mass transfer rate from columnar to equiaxed due to capillary-driven fragmentation. $d_{e,frag}^0$: initial size of the fragment. $S_V$: interfacial area density of the fragment. |
| $S_V = f_c(1-f_c)^{\bar{r}}((S_{S0}^{-1})^3 + K_0 t_f)^{-1/3}$ | (6) | |
| $d_{e,frag}^0 = f_c \frac{1.6}{S_V}$ | (7) | |
| $M_{frag}^{capillary} = N_{frag}^{capillary}(\rho_e \frac{\pi}{6}(d_{e,frag}^0)^3)$ | (8) | |

simulation and experiment was achieved in terms of the distributions of both spurious grains and freckles.

It is essential to differentiate between freckles and channel segregation. A segregation channel can only be termed a freckle when it has entrapped spurious grains, whereas a segregation channel without spurious grains can at most be named 'quasi-freckle'. The extended digital twin method provides insights into the origin of spurious grains and freckles. It is channel segregation that initiates spurious grains, leading to the formation of freckles, rather than spurious grains initiating channels and freckles. In other words, freckles originate from the segregation channels.

The digital twin method with a three-phase solidification model has been proven to be applicable for engineering turbine blades. However, its computational cost remains very high. The current study also shows that a two-phase model that neglects the formation of spurious grains can provide qualitative solutions for channel segregation while being economical. However, a three-phase solidification model with crystal fragmentation must be considered if quantitative results regarding freckles are of primary interest.

## Methods
A two-step digital twin approach[25] was taken. This was achieved by coupling ProCAST with a volume-average solidification model. The global thermal field in the Bridgman furnace, including the layout of the mould and casting system, was calculated using ProCAST. Flow and solidification within the casting were calculated using a volume-average solidification model[21,36]. The casting surface temperature profiles calculated by ProCAST were output and stored as $T$-$t$ curves, which were then applied as the Dirichlet thermal boundary condition for the flow-solidification simulation. In what follows, a brief overview of the volume-average solidification model is presented, followed by a detailed interpretation of the model refinements and simulation settings, and concluding with an introduction to the directional solidification experiment.

### Brief description of volume-average solidification model
The previously developed mixed-columnar-equiaxed volume-average solidification model[21,36] was extended and used to solve the flow and solidification problems in superalloy casting. The key features of this model are as follows:

Three hydrodynamic phases are involved: liquid melt, equiaxed (spurious) grains, and columnar dendrites with volume fractions that total one ($f_\ell + f_e + f_c = 1$). To calculate the solidification/remelting rate,

simplified crystal morphologies are assumed, that is, stepwise cylinders for columnar and spheres for equiaxed. Note that this assumption does not apply to the sub-models for fragmentation, where dendrite structures are considered. Thermodynamic equilibrium is applied at the solid–liquid interface. Diffusion-governed growth kinetics is considered, and the difference between the thermodynamic equilibrium concentration of the liquid at the interface ($c_\ell^*$) and the volume-averaged liquid concentration ($c_\ell$) serves as the driving force for solidification and remelting. The columnar structure is initiated only from the bottom of the casting, and its tip front is traced using the Lipton–Glicksman–Kurz (LGK) model[43]. The mushy zone is treated as a porous medium by solving Darcy's law. The thermal–solutal convection is approximated using the Boussinesq approach. The solidification-shrinkage-induced feeding flow is considered. Spurious grains are assumed to originate only from dendrite fragmentation. Remelting and destruction of equiaxed grains are also considered. Macrosegregation is characterised by the segregation index $c_{mix}^{index} = 100 \cdot (c_{mix} - c_0)/c_0$. Further details regarding the volume-average solidification model are provided in previous studies[21,36,44]. Only the modelling parts related to the fragmentation and remelting/grain-destruction sub-models are detailed below. The conservation of the number density of equiaxed grains is expressed by Eq. (1) in Table 1, where the different source/sink terms represent the origin of the equiaxed grains (or crystal fragments) by different fragmentation mechanisms or crystal destruction owing to remelting. Some of these terms are listed in Table 1, and further explanations are provided below.

### Grain destruction by remelting
The sizes of the equiaxed grains are assumed to follow a lognormal size distribution, as shown in Eq. (2). If equiaxed grains are exposed to a superheated liquid, the remelting process begins with a reduction in the grain diameter. The remelting rate of the grains, $d(d_e)/d(t)$, yields Eq. (3). The effect of liquid convection on the remelting rate is considered by adjusting the diffusion length[45]. When the grain diameter remelts below a critical value ($d_{e,critical}$), the grain is eliminated from the system; that is, grain destruction occurs. The destruction rate of grain number density is calculated using Eq. (4)[46].

### Capillary-driven fragmentation
Based on a recent microgravity solidification experiment aboard the International Space Station[31], a capillary-driven fragmentation model was

**Fig. 10 | Schematic of the dynamic behaviour of the flow-driven fragmentation process. a** A columnar dendrite before fragmentation. The green colour indicates the solute concentration, and the red arrows indicate the flow direction. **b1, b2** The morphological changes in the side arms before and after pinch-off. **b3** The $t_r - \tau$ criterion for the timing of dendrite pinch-off by flow-driven fragmentation. **c** The neck diameter of side arm roots ($d_{pr}$) as a linear function of $(t - t_s)^{1/3}$.

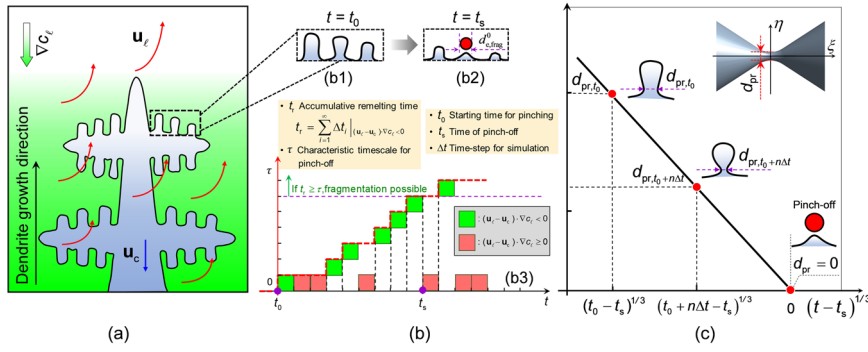

developed as represented by Eq. (5)[33]. The capillary-driven fragmentation rate ($N_{frag}^{capillary}$) is equal to the time derivative of the cube of $S_V$ multiplied by the constant $a$, where $a$ is a material-dependent constant and $S_V$ is the interfacial area density, which can be calculated according to the coarsening law (Eq. (6))[47]. The initial diameter of the formed fragment $d_{e,frag}^0$ is estimated using Eq. (7)[48]. Assuming a spherical morphology for the fragments, the mass transfer between the columnar and equiaxed phases ($M_{frag}^{capillary}$) owing to capillary-driven fragmentation yields Eq. (8).

**Flow-driven fragmentation model**
Based on Flemings' theory, which states that interdendritic flow in the growth direction of primary columnar dendrites promotes remelting[28,49], a flow-driven fragmentation model was previously proposed[32]. Flow-driven remelting occurs once the $(\mathbf{u}_\ell - \mathbf{u}_c) \cdot \nabla c_\ell < 0$ criterion is fulfilled, in which $\mathbf{u}_\ell$ and $\mathbf{u}_c$ are the velocities of the liquid and columnar phases, respectively, and $\nabla c_\ell$ is the liquid concentration gradient. The original models ignored the timing of fragmentation, i.e., the dendrite pinch-off. The fragmentation was assumed to occur either at a depth greater than eight times the secondary dendrite arm spacing deep into the columnar mushy zone[28] or at a volume fraction ($f_c$) higher than 0.03 in the columnar phase[32], when the above remelting criterion is also fulfilled. During the solidification process, fragmentation does not occur until the roots of the high-order side arms are remelted off from the main stem. This remelting process involves a kinetic pinching process[47,50]. As shown schematically in Fig. 10a, the liquid flows in the growth direction of the primary dendrite, which fulfils the above remelting criterion. The morphological changes in the side arms before and after pinch-off by remelting are illustrated in the insets of Fig. 10b1, b2. If the dendrite root detaches from the main dendrite stem before the head melts completely, one fragment is produced.

A recent theoretical and experimental investigation suggests that the pinching process follows a self-similar interfacial evolution[40]. As shown schematically in Fig. 10c, the diameter of the pinching rod ($d_{pr}$) decreases with $(t - t_s)^{1/3}$. Here, $t$ is the time, and $t_s$ is the pinch-off time. As illustrated in Fig. 10b3, c, the dendrite root starts to pinch at moment $t_0$ with an initial diameter of $d_{pr,t_0}$. By projecting the pinching side arm onto a time-independent cylindrical coordinate, with the axis of the side arm coincident with the $\xi$-axis, as shown by the inset in the upper right corner in Fig. 10c, the interface position can be described by the function $\eta = f(\xi)$. The diameter at the centre ($\xi = 0$) of the pinching side arm evolves according to Eq. (9)[40].

$$d_{pr}(t) = 2f(0)[D_\ell \lambda_0 (t_s - t)]^{1/3} \quad (9)$$

where $f(0)$ is the interface position at $\xi = 0$, $D_\ell$ is the diffusion coefficient of the solute in the liquid ($D_\ell = 3.6 \times 10^{-9} \, m^2 s^{-1}$)[22], and $\lambda_0$ is the capillary length of the solid–liquid interface ($\lambda_0 = 2.27$ nm)[40]. According to Aagesen et al.[40], a value of 0.6 is used for $f(0)$. The characteristic time ($\tau$) for pinch-off, i.e., from the start of pinching to pinch-off, is equal to $t_s - t_0$. From Eq. (9), $\tau$ can be derived and is expressed as shown in Eq. (10). Owing to the lack of relevant data, $d_{pr,t_0}$ was determined through test simulations by fitting the

simulation results to the experiments.

$$\tau = t_s - t_0 = \frac{d_{pr,t_0}^3}{8f(0)^3 D_\ell \lambda_0} \quad (10)$$

Here, a $t_r - \tau$ criterion is suggested for the timing of dendrite pinch-off by flow-driven fragmentation, where $t_r$ is the accumulative remelting time (i.e., $t_r = \sum_{i=1}^n \Delta t_i |_{(\mathbf{u}_\ell - \mathbf{u}_c)\nabla c_\ell < 0}$) numerically recorded from time $t_0$. The recorded $t_r$ was then compared with $\tau$. As illustrated in Fig. 10b3, only if $t_r > \tau$, fragmentation is possible. The Heaviside step function ($H_x$)[51] was employed to describe the possibility of fragmentation, as shown in Eq. (11), where $\tilde{k}$ ($\tilde{k} = 200$) is the Heaviside step function constant.

$$H_x = \left\{ 1 + \exp\left[2\tilde{k}\left(1 - \frac{t_r}{\tau}\right)\right] \right\}^{-1} \quad (11)$$

Given the above considerations, the flow-driven dendrite fragmentation model is refined to consider the pinch-off dynamics of the side arms. The mass transfer rate from columnar to equiaxed ($M_{frag}^{flow}$) owing to flow-driven fragmentation is modified as Eq. (12).

$$M_{frag}^{flow} = -\gamma \cdot H_x \cdot (\mathbf{u}_\ell - \mathbf{u}_c) \cdot \nabla c_\ell \cdot \rho_e \cdot f_c \quad (12)$$

where $\mathbf{u}_\ell$ and $\mathbf{u}_c$ are the velocities of the liquid and columnar phases, respectively, $\nabla c_\ell$ is the liquid concentration gradient, $\rho_e$ is the density of the equiaxed phase, $f_c$ is the volume fraction of the columnar phase, and $\gamma$ is the fragmentation constant. Because of the assumed spherical morphology of the produced fragments, the production rate of the flow-driven fragment $N_{frag}^{flow}$ yields Eq. (13), where $d_{e,frag}^0$ is the initial fragment diameter, which can be calculated using Eqs. (6) and (7).

$$N_{frag}^{flow} = \frac{M_{frag}^{flow}}{\frac{\pi}{6}(d_{e,frag}^0)^3 \rho_e} \quad (13)$$

**Simulation settings**
The DS of a superalloy casting was simulated. Thermal calculations were conducted in ProCAST using the full-scale geometry shown in Fig. 11b. The temperatures of the top hot chamber and bottom cold chamber were set to 1750 and 353 K, respectively. Radiation was considered between the inducting wall and shell mould. The casting system was then withdrawn by raising the furnace at a speed of 1.5 mm/min. The calculation started with a mould fully filled with hot melt (1750 K); that is, the filling process was ignored. The stabilization time following pouring was not counted in the current simulation. The conservation of enthalpy was solved for the shell mould and casting, but only the latent heat due to the solidification of the casting was treated with an equivalent specific heat method.

The flow-solidification calculation using the volume-average model was limited to the casting body, as illustrated in Fig. 11c. The cooling history

**Fig. 11 | Overview of the Bridgman-type vacuum furnace and casting system. a** Configuration of the industial Bridman-type vacuum furnace and casting system. **b** A layout of one-twelfth of the furnace and the casting system. **c** The geometry and height of the casting. **d** The cross-section of the casting. **e** Assembly of the casting in the Bridgman furnace.

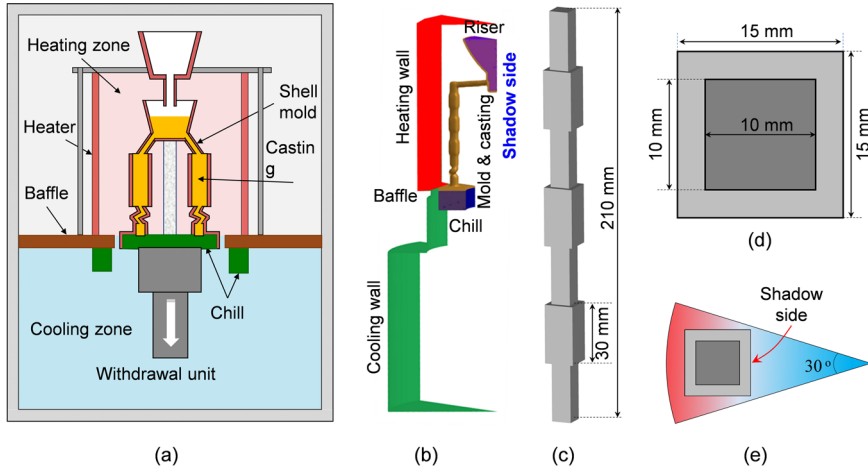

## Table 2 | Material properties and processing parameters

| Properties/parameters | Symbol | Units | Values | Refs |
|---|---|---|---|---|
| **Thermophysical** | | | | |
| Diffusion coefficient in liquid | $D_\ell$ | m²·s⁻¹ | $3.6 \times 10^{-9}$ | 22 |
| Density of liquid | $\rho_\ell$ | kg·m⁻³ | 7646.0 | 52 |
| Density of solid (equiaxed and columnar) | $\rho_e, \rho_c$ | kg·m⁻³ | 8278.0 | 53 |
| Thermal conductivity of liquid | $k_\ell$ | W·m⁻¹·K⁻¹ | 24.6 | 52,54 |
| Thermal conductivity of solid | $k_e, k_c$ | W·m⁻¹·K⁻¹ | 33.5 | 52,54 |
| Specific heat | $c_p^\ell, c_p^e, c_p^c$ | J·kg⁻¹·K⁻¹ | 575.0 | 38,52 |
| Latent heat of fusion | $\Delta h_f$ | J·kg⁻¹ | $2.4 \times 10^5$ | 38,55 |
| Thermal expansion coefficient | $\beta_T$ | K⁻¹ | $-1.16 \times 10^{-4}$ | 22 |
| Solutal expansion coefficient | $\beta_c$ | - | −0.228 | 22,23,56 |
| Viscosity | $\mu_\ell$ | kg·m⁻¹·s⁻¹ | $4.9 \times 10^{-3}$ | 22 |
| **Thermodynamic** | | | | |
| Equivalent initial concentration | $c_0$ | - | 0.3509 | 25,42 |
| Equivalent liquidus slope | $m$ | K (wt.%)⁻¹ | −1.145 | 23,56 |
| Equivalent equilibrium partition coefficient | $k$ | - | 0.57 | 56–58 |
| Initial liquidus | $T_L$ | K | 1688.0 | - |
| Eutectic temperature | $T_{eut}$ | K | 1627.0 | - |
| Primary dendritic arm spacing | $\lambda_1$ | μm | 300.0 | Exp. measurement |
| Gibbs–Thomson coefficient | $\Gamma$ | m·K | $3.6 \times 10^{-7}$ | 59 |
| Melting point of the solvent | $T_f$ | K | 1728.0 | - |
| **Fragmentation parameters** | | | | |
| Coarsening constants | $\tilde{r}$ | - | 0.4 | 47 |
| | $S_{S0}^{-1}$ | μm | 246.0 | - |
| | $K_0$ | μm³·s⁻¹ | 23.5 | 47 |
| Flow-driven fragmentation coefficient | $\gamma$ | - | $5.0 \times 10^{-4}$ | - |
| Capillary-driven fragmentation coefficient | $a$ | - | $1.0 \times 10^{-5}$ | - |
| Heaviside step function constant | $\tilde{k}$ | - | 200.0 | - |
| Capillary length for the solid-liquid interface | $\lambda_0$ | m | $2.27 \times 10^{-9}$ | 40 |
| Interface position of pinching root at $\xi = 0$ | $f(0)$ | - | 0.6 | 40 |

of the temperature distribution on the casting surface, calculated using ProCAST, was used as the Dirichlet thermal boundary condition for the flow and solidification calculation. A no-slip flow boundary condition was applied to the lateral wall of the sample, and a pressure inlet flow boundary condition was applied to the top surface. The multicomponent superalloy was simplified as a binary alloy (Ni-35.09 wt.% solute) with constant liquidus slope and solute partition coefficient[42]. The alloy was assumed to be incompressible with a constant viscosity. The material properties and other parameters are listed in Table 2. The volume-average multiphase solidification model was implemented on the platform of Ansys Fluent 17.1 version. The casting was meshed into 260,000 cubic cells with a mesh size of 0.5 mm. Using a time step of 0.01 s, one simulation took approximately 10 days on a high-performance cluster (2.6 GHz and 28 cores).

**Table 3 | Main composition of the superalloy**

| Elements | Ta | Cr | Al | Co | W | Re | Ti | Mo | Ni |
|---|---|---|---|---|---|---|---|---|---|
| Content (wt.%) | 8.07 | 3.39 | 5.69 | 5.97 | 6.52 | 4.89 | 0.15 | 0.41 | Balance |

### Directional solidification experiments

The experiments were conducted using DS in an industrial-scale Bridgman-type vacuum furnace. As shown schematically in Fig. 11a, the furnace includes a hot chamber at the top, a cold chamber at the bottom, and a baffle in the middle. A cluster of superalloy castings was assembled around the axis of the furnace, but here only the layouts of one-twelfth of the furnace and the casting system are shown in Fig. 11b. Ceramic shell moulds were obtained using a standard lost-wax procedure. A typical spiral grain selector including a starter block was used to achieve single-crystal growth. The moulds were preheated to 1750 K in the furnace, and the hot melt (1750 K) was then poured into the moulds. After approximately 10 min of stabilisation, solidification of the casting was triggered by withdrawing the casting downward from the heated zone to the cold zone. A low withdrawal velocity of 1.5 mm/min was intentionally used to create a freckle-prone condition. The shape and dimensions of the casting are illustrated in Fig. 11c, d. The casting is 210 mm high and consists of seven segments of equal height with different cross-section sizes. Both the small and large cross-sections have a square shape, and the side lengths are 10 mm and 15 mm, respectively. The assembly of castings in the furnace is shown in Fig. 11e. The compositions of the superalloys are listed in Table 3.

### Data availability

The data that support the findings of this study are available from the corresponding authors upon reasonable request.

### Code availability

The code that supports the findings of this study is available from the corresponding authors upon reasonable request.

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

## Acknowledgements

This work was financially supported by the Austrian Science Fund in the framework of the FWF-NKFIN joint project (FWF, I4278-N36).

## Author contributions

H.Z. and M.W. proposed the original research concept and developed the methodology. H.Z. performed the flow and solidification modelling and W.X. performed the thermal calculation. D.M. and Y.Z. conducted the experimental investigations. H.Z. and M.W. analyzed the results and drafted the manuscript. A.L., D.M., and A.K. reviewed and discussed the manuscript. The project was coordinated by M.W.

## Competing interests
The authors declare no competing interests.
