## [Transparent Peer Review file · Communications Materials]

Modelling freckles and spurious grain formation in directionally solidified superalloy castings

Corresponding Author: Professor Menghuai Wu

Version 0:

Decision Letter:

Dear Professor Wu,

Thank you for submitting your manuscript, "Modelling study of the freckles and spurious grain formation in directionally solidified superalloy castings", to Communications Materials. It has now been seen by 3 referees, whose comments are appended below. You will see that while they find your work of interest, some important points are raised. We are interested in the possibility of publishing your study in Communications Materials, but would like to consider your response to these concerns in the form of a revised manuscript before we make a decision on publication.

We therefore invite you to revise and resubmit your manuscript, taking into account the points raised.

When submitting your revised manuscript, please include the following:

-A response letter with a point-by-point reply to each of the referee comments and a description of changes made. Please include the complete referee report in the response letter. Please note that the response letter must be separate to the cover letter to the editors.

-A marked-up version of the manuscript with all changes to the text in a different colored font. Please do not include tracked changes or comments. Please select the file type 'Revised Manuscript - Marked Up' when uploading the manuscript file to our online system.

-A clean version of the manuscript. Please select the file type 'Article File'.

-An updated [Editorial Policy](https://www.nature.com/documents/nr-editorial-policy-checklist.zip) checklist, uploaded as a 'Related Manuscript File' type. This checklist is to ensure your paper complies with all relevant editorial policies. If needed, please revise your manuscript in response to these points. Please note that this form is a dynamic 'smart pdf' and must therefore be downloaded and completed in Adobe Reader. Clicking this link will download a zip file containing the pdf.

In the event that your manuscript is accepted we will provide detailed guidance on our journal policies and formatting. You may however wish to ensure that the manuscript complies with our house style at this stage. See our style and formatting guide (<https://www.nature.com/documents/commsj-phys-style-formatting-guide-accept.pdf>) and checklist (<https://www.nature.com/documents/commsj-phys-style-formatting-checklist-article.pdf>) for reference.

Data availability statements and data citations policy: All Communications Materials manuscripts must include a section titled "Data Availability" at the end of the Methods section or main text (if no Methods). More information on this policy, and a list of examples, is available at <http://www.nature.com/authors/policies/data/data-availability-statements-data-citations.pdf>.

- Accession codes for deposited data
- Other unique identifiers (such as DOIs and hyperlinks for any other datasets)
- At a minimum, a statement confirming that all relevant data are available from the authors

- If applicable, a statement regarding data available with restrictions
- If a dataset has a Digital Object Identifier (DOI) as its unique identifier, we strongly encourage including this in the Reference list and citing the dataset in the Data Availability Statement.

DATA SOURCES: We strongly encourage authors to deposit all new data associated with the paper in a persistent repository where they can be freely and enduringly accessed. We recommend submitting the data to discipline-specific, community-recognized repositories, where possible and a list of recommended repositories is provided at <http://www.nature.com/sdata/policies/repositories>.

If a community resource is unavailable, data can be submitted to generalist repositories such as [figshare](https://figshare.com/) or [Dryad Digital Repository](http://datadryad.org/). Please provide a unique identifier for the data (for example a DOI or a permanent URL) in the data availability statement, if possible. If the repository does not provide identifiers, we encourage authors to supply the search terms that will return the data. For data that have been obtained from publically available sources, please provide a URL and the specific data product name in the data availability statement. Data with a DOI should be further cited in the methods reference section.

Please use the following link to submit your documents:

Link Redacted

We hope to receive your revised paper within three months; please let us know if you aren't able to submit it within this time so that we can discuss how best to proceed. If we don't hear from you, and the revision process takes significantly longer, we will close your file. In this event, we will still be happy to reconsider your paper at a later date, as long as nothing similar has been accepted for publication at Communications Materials or published elsewhere in the meantime.

Please do not hesitate to contact me if you have any questions or would like to discuss these revisions further. We look forward to seeing the revised manuscript and thank you for the opportunity to review your work.

Best regards,

Xiaoyan Li, PhD
Editorial Board Member
Communications Materials
orcid.org/0000-0002-2953-9267

Reviewers' comments:

Reviewer #1 (Remarks to the Author):

The author has achieved good results in simulation of freckle-formation by considering spurious grain in freckles and using a self-designed simulation approach based on the volume-average method. Some problems as following should be modified before acceptance.

Suggestions:

- (1) A recent research shows (10.1016/j.actamat.2024.119702) that the Ra criterion can be upgraded considering the geometrical effects to complex-shaped cases, so the description of "The Ra criterion fails to consider..." in the introduction section is seemed to be too absolute.
- (2) Too long to correctly understand the sentence "Diffusion of the solute in the liquid leads to..." for the readers. Please simplify it.
- (3) It is recommended a reference here at "The Heaviside step function(Hx) was employed..."
- (4) It is recommended to define these symbols where first introduced in the ahead section. "where \bar{u}_l and \bar{u}_c are the velocities of the liquid and columnar phases, respectively..."
- (5) What is the relationship between the defined $d_{e,grag_0}$ and $d_{pr,t0}$?
- (6) In Figs. 5-11, it is necessary to add the heater-shadow directions.
- (7) In section 5.1, the morphology of sliver is usually regarded different from the freckle in general, please give a simple explanation after the description "if the current model is to be applied to model sliver defects"
- (8) In section 5.2, a simple description should be added after the sentence "The current modelling tool provides..." to declare which geometrical effect can be quantified and how it works.
- (9) Some symbols in section 5.3 are not displaying correctly, please check the resource file format.

Questions to answer:

- (1) The concept of digital-twin is more inclined towards a digital object related to REAL-TIME interactive data (not only by

simulation). That is, the data simulated should be connected with the data from the sensors. The author should pay attention to the differences between digital twin and simulation, or give an extra explanation when used.

(2) For all the problems considering flow field details (such as the cracking of SDA under the influence of flow), the DIRECTION distribution of the flow field is crucial. However, significant flow field deviations may occur due to numerical algorithm issues in the CFD process. For example, the U-xyz distributions obtained using RANS, LES and DNS will inevitably be different, for typical descriptions of convective turbulence. Therefore, how to ensure the correctness of the simplified flow field when it interacts with the microscale SDA?

(3) The author's main focus is on spurious grain in freckles, but the impact degree appears to be limited based on the final results. However, it is still necessary to quantify the contribution of spurious grain to freckles and provide an estimation in the discussion section, based on the current simulation results.

(4) The relationship between eutectic and freckle is closely related, based on the existing simulation results. However, these relationships are embedded within the text and not explicitly expressed. Could the author provide a clear and concise explanation of this connection based on the available results?

Reviewer #2 (Remarks to the Author):

Single-crystal superalloy turbine blades (TBs) are susceptible to freckle defects. Despite many studies were made on this topic, the formation of spurious grains and their relationship to the segregation channel are not well understood. This article deals with the modelling of the formation of segregation channels and spurious grains during the directional solidification of superalloy castings. A three-phase mixed columnar-equiaxed solidification model, considering the interaction between the grain motion and liquid flow within the segregation channels, was used to model the solidification process. Two crystal fragmentation theories were integrated into the model, and a τ - τ criterion was proposed to treat the timing of dendrite pinch-off by the flow-driven fragmentation mechanism. Directional solidification experiments for superalloy castings were conducted. The modelling results were well validated by the experiments in terms of the distributions of both spurious grains and freckles.

This article is of interest, and the approach proposed by the authors is clearly relevant. The formation of spurious grains via the fragmentation mechanism within a developed segregation channel is verified. The evolution of an as-developed freckle passing through a cross-sectional change is demonstrated. The application of the extended model to an engineering turbine blade is also interesting. The initiation of a freckle from an as-developed segregation channel is also confirmed. Such results extend the understanding on the formation of freckles and the associated spurious grains.

To the best of my knowledge, the results presented in this paper represent the state-of-the-art in modelling freckles and spurious grains in superalloy turbine blades. In my opinion, the article is worthy of acceptance by Communications Materials after some minor revisions. Consequently, I have the following remarks:

1. Page 10: Why was the Heaviside step function chosen to describe the probability of fragmentation?
2. Figure 7: The results show that the number of fragments from flow-driven fragmentation is much greater than that from capillary-driven fragmentation. However, previous studies indicate that the primary source of dendrite fragmentation is capillary-driven (T. Cool, *Acta Mater.*, 2017).
3. Figure 9: Considering the formation of spurious grains seems to have no impact on the predicted distribution of freckles. Does this mean that spurious grains can be ignored in future freckle simulations?
4. The calculation of a single case took 10 days on a high-performance cluster. With the advancement of computer technology, this calculation time may be reduced in the future. In addition to improvements in computational capacity, are there any possibilities to enhance the entire VA-model algorithm to accelerate the calculation?

Reviewer #3 (Remarks to the Author):

In the submitted paper, the authors present a method for predicting the formation of spurious grains and segregation channels in turbine blades, developed on the basis of their earlier work [*Acta Mater* 244, 2023, 118579]. In this current research, the authors developed a mechanism of columnar dendrite fragmentation which they then implemented into the digital twin method to simulate the formation of spurious grains in the segregation channel during directional solidification of castings. A sensible classification has been made on freckles, spurious grains and segregation channels in this article. Freckles are a serious problem during directional solidification of single crystal nickel superalloy castings. Sometimes the reason of their formation is very difficult to explain especially in complex shaped blades. The results presented in this paper show that for a changing casting cross-section, the fluid flow above the solidification front and inside the mushy zone, which determines the formation of the segregation channel and spurious grains, is particularly difficult to determine. In my opinion, the presented results can be very interesting and helpful for people working in the production and development of single crystal castings. The academic community can also use the results to interpret their studies of freckle formation, segregation channels and fluid flow in directionally solidified castings. The authors of this paper have a great knowledge of solidification and also the manufacturing of single crystals. The paper is well written and organized. In my humble opinion, this paper should be accepted for publication after a minor revision. I have included some comments and questions below:

Minor remarks:

1. Line 108, in the text is "central rod" while it is not included in the model presented in figure 1b. Also, there is a hole in the chill plate and mold. Why was the central rod removed and the hole in the chill plate included in the simulation model ?
2. Line 113, the temperature of the melt poured into the mold cavity should be given.

3. It would be good to show the shadow side on the drawing of the furnace (Fig 1).
4. Line 148. "(d1)-(d3) Microstructure on transverse sections". Microstructures (d1) and (d2) are not included in Fig. 2 . Please correct this.
5. The two areas marked with red dashed lines in Fig 2c are not described in the text (segregation channels ?).
6. According to the description in the text, 12 castings were manufactured. What was the basis for selecting this one representative casting shown in Fig. 2?
7. Line 202, "uc" not added in Fig. 3a.
8. Better modify Fig. 3a to show directional solidification in a vertical configuration and columnar dendrites, which are more consistent with experiment and simulations for the Bridgman process.
9. Line 311, from which area is the solute-enriched liquid sucked into the channels, the interdendritic area?

Major remarks:

1. Most of the simulations were carried out for the lower segments of the casting, whereas the microstructure of the freckles and channels is only shown for the top segment. It would also be good to show the microstructure of the freckles on the surface of the lower segment (lower casting area) in order to compare experimental and simulated results. I think this would be more interesting for the reader. The morphology of the freckles in the lower segment of the casting is very poorly visible in Fig. 1a.
2. According to Fig.4c, for a time of 4000 s, the temperature at points A and B reaches a value of about 1650 K, whereas the temperature value in Fig.4b for these points is much less than 1600 K (same time of 4000 s). These values are not consistent in the two figures for the same times. Please check this.
3. In my opinion, it should be emphasized in the text that the solidification times given for the simulations shown in the Figs. 5 and 6 do not take into account the solidification stage of the starter and selector. This can be confusing for the reader, especially since simulation results (Fig 4a) obtained in ProCAST software are also shown in the paper.
4. Line 396 "The upper part of the casting was more susceptible to freckles than the lower part ". It would be interesting for readers to explain in more detail in the paper why the propensity to freckle formation is higher in the upper part of the casting than in the lower part. From my observations it seems that the curvature of the liquidus isotherm is usually highest at the top of the casting. Perhaps this is related to the freckle formation.

Communications Materials is committed to improving transparency in authorship. As part of our efforts in this direction, we are now requesting that all authors identified as 'corresponding author' create and link their Open Researcher and Contributor Identifier (ORCID) with their account on the Manuscript Tracking System prior to acceptance. ORCID helps the scientific community achieve unambiguous attribution of all scholarly contributions. You can create and link your ORCID from the home page of the Manuscript Tracking System by clicking on 'Modify my Springer Nature account' and following the instructions in the link below. Please also inform all co-authors that they can add their ORCIDs to their accounts and that they must do so prior to acceptance.

Version 1:

Decision Letter:

Dear Professor Wu,

Your manuscript titled "Modelling study of the freckles and spurious grain formation in directionally solidified superalloy castings" has now been seen again by Reviewer 2 and 3, whose comments appear below. Reviewer 1 did not respond to our requests, so we checked your replies to them ourselves. I am delighted to say that we are happy, in principle, to publish a suitably revised version in Communications Materials.

We therefore invite you to edit your manuscript to comply with our journal policies and formatting style in order to maximise the accessibility and therefore the impact of your work.

EDITORIAL REQUESTS

* Your manuscript should comply with our policies and format requirements, detailed in our style and formatting guide (<https://www.nature.com/documents/commsj-phys-style-formatting-guide-accept.pdf>).

* Please edit your manuscript according to the editorial requests in the attached table, and outline revisions made in the right hand column. If you have any questions or concerns about any of our requests, please do not hesitate to contact me. It is important that each request be addressed in order to avoid delays in accepting your manuscript. Please upload the completed table with your manuscript files as a Related Manuscript file.

* The editorial requests table also includes a full list of the files that must be provided upon resubmission. Please upload your files according to this table.

* An updated editorial policy checklist that verifies compliance with all required editorial policies must be completed and uploaded with the revised manuscript. All points on the policy checklist must be addressed; if needed, please revise your manuscript in response to these points. Please note that this form is a dynamic 'smart pdf' and must therefore be downloaded and completed in Adobe Reader. Clicking this link will download a zip file containing the pdf.

OPEN ACCESS

Communications Materials is a fully open access journal. Articles are made freely accessible on publication. For further information about article processing charges, open access funding, and advice and support from Nature Research, please visit <https://www.nature.com/commsmat/open-access>

Please use the following link to submit your revised files:

Link Redacted

We hope to hear from you within two weeks; please let us know if the process may take longer.

Best regards,

Xiaoyan Li, PhD
Editorial Board Member
Communications Materials
orcid.org/0000-0002-2953-9267

REVIEWERS' COMMENTS:

Reviewer #2 (Remarks to the Author):

The authors have addressed all the comments properly, therefore it is suitable for publication.

Reviewer #3 (Remarks to the Author):

The authors have addressed all my comments. In my opinion, this paper should be accepted for publication.

Point-to-point response to the reviewer and editor

Thank you very much for your kind comments on this paper. We have revised the manuscript and highlighted the changes in red. The following is the point-to-point response.

Reviewer #1:

Suggestions:

(1) A recent research shows (10.1016/j.actamat.2024.119702) that the Ra criterion can be upgraded considering the geometrical effects to complex-shaped cases, so the description of “The Ra criterion fails to consider...” in the introduction section is seemed to be too absolute.

Answer: Thank you very much for your comments. We do not think Ra criterion can predict freckles, because the initiation and development of freckles are related to fluid dynamics and its interaction with the advancing mushy zone. For the superalloy turbine blades with complex outer and inner structures, the flow dynamics are extremely chaotic. Without rigorous flow and solidification calculations, it is impossible to quantitatively predict freckle formation. However, we respect the reviewer's comment and have removed the sentence in the revised manuscript on Page 2.

(2) Too long to correctly understand the sentence “Diffusion of the solute in the liquid leads to...” for the readers. Please simplify it.

Answer: Yes! It was simplified.

Page 3: “Diffusion of the solute in the liquid leads to dendrite growth in regions of lower curvature. This occurs at the expense of regions with higher curvature.”

(3) It is recommended a reference here at “The Heaviside step function($H(x)$) was employed...”

Answer: Yes! A reference was added.

Page 10: “The Heaviside step function ($H(x)$) [47] was employed ...”

(4) It is recommended to define these symbols where first introduced in the ahead section. “where \bar{u}_l and \bar{u}_c are the velocities of the liquid and columnar phases, respectively...”

Answer: Yes! The original paper was revised on Page 9.

Page 9: “Flow-driven remelting occurs once the $(\bar{u}_l - \bar{u}_c) \cdot \nabla c_l < 0$ criterion is fulfilled, in which \bar{u}_l and \bar{u}_c are the velocities of the liquid and columnar phases, respectively, and ∇c_l is the liquid concentration gradient.”

(5) What is the relationship between the defined $d_{e,grag_0}$ and $d_{pr,t0}$?

Answer: Thank you very much for your comment! $d_{pr,t0}$ is the initial diameter at the center of a pinching side arm, while $d_{e,frag}^0$ is the volume-averaged diameter of the fragmented side arms, i.e. the initial diameter of the produced fragments. They do not have a direct relationship.

(6) In Figs. 5-11, it is necessary to add the heater-shadow directions.

Answer: Yes! All figures have been updated. Because the heated surface and the shadow surface are two opposite surfaces, only the shadow surface is marked.

(7) In section 5.1, the morphology of sliver is usually regarded different from the freckle in general, please give a simple explanation after the description “if the current model is to be applied to model sliver defects”

Answer: We agree with you that the morphology of the sliver is usually regarded differently from the freckle. The freckle is identified as a thin chain of small misoriented (big angel difference) grains associated with local accumulation of eutectics. The sliver is generally characterized as a thin and long misoriented (small angel difference) grain extending along the solidification direction. Although some tiny spurious grains within the freckles may develop into long sliver defects, this is not the primary mechanism behind sliver defect formation.

After carefully thinking, we think it is too ambitious to model the sliver defect using the current model. Therefore, we removed this sentence in the revised manuscript on Page 19.

(8) In section 5.2, a simple description should be added after the sentence “The current modelling tool provides...” to declare which geometrical effect can be quantified and how it works.

Answer: Yes! The original text has been reorganized.

Page 21: “The geometrical effect on freckle formation in superalloy castings has been experimentally demonstrated [13-15], but the complex interplay between the thermo-solutal convection and feeding flow was not explained. The current modelling tool provides a quantitative explanation to the geometrical effect.”

(9) Some symbols in section 5.3 are not displaying correctly, please check the resource file format.

Answer: Checked! The incorrect display of some symbols may be caused by the system when converting the file to PDF

Questions to answer:

(1) The concept of digital-twin is more inclined towards a digital object related to REAL-TIME interactive

data (not only by simulation). That is, the data simulated should be connected with the data from the sensors. The author should pay attention to the differences between digital twin and simulation, or give an extra explanation when used.

Answer: Thank you very much for your comments! We agree with you. Achieving a true digital twin remains a significant and long-term research challenge. As the first but important step, we developed a numerical tool, which can provide a digital replica of the real process. Online and real time interaction with the physical process is still impossible currently. To avoid any misunderstanding to the potential reader, new statement has been added in the revised manuscript.

Page 3: “Although real-time interaction with the physical process remains unattainable, it is possible to create a digital replica of the physical process.”

(2) For all the problems considering flow field details (such as the cracking of SDA under the influence of flow), the DIRECTION distribution of the flow field is crucial. However, significant flow field deviations may occur due to numerical algorithm issues in the CFD process. For example, the U-xyz distributions obtained using RANS, LES and DNS will inevitably be different, for typical descriptions of convective turbulence. Therefore, how to ensure the correctness of the simplified flow field when it interacts with the microscale SDA?

Answer: You are absolutely correct that the DIRECTION distribution of the flow field is crucial to dendrite fragmentation. For the current study, because the flow belongs to laminar flow (Reynold number<50), it is not necessary to use any turbulent model like RANS or LES. Any uncertainties associated with the turbulent model have been avoided. A coarse-mesh-DNS model was used in this study with a relatively fine mesh size (500 μm). We have done a mesh sensitivity study, and the modeling results demonstrate that the currently used mesh size is fine enough to ‘reproduce the development of the freckles during the directional solidification process. The satisfactory agreement between the modeling results and the experimental measurement in this work proves indirectly the correctness of the calculated flow field.

In a solidification model, how to model the turbulence effects in the mushy zone (mixed solid-liquid region) is still very difficult. When turbulence is considered, a critical point is to properly treat the turbulence-induced energy and species diffusive terms. To the current authors’ knowledge, there is no effective way to model such terms. Without rigorous derivation and careful treatment, it can easily over- or under-estimate the turbulence effect on the species transport near the solidification front, and then lead to unacceptable calculation results. Somehow, we do not trust any turbulent model in multiphase solidification modeling.

(3) The author's main focus is on spurious grain in freckles, but the impact degree appears to be limited based on the final results. However, it is still necessary to quantify the contribution of spurious grain to

freckles and provide an estimation in the discussion section, based on the current simulation results.

Answer: Thank you very much for your comments! Considering the spurious grains has very limited effect on the sample with cross-section change, but it has significant effect on the industrial-scale turbine blade that was modelled in section 5.4. New discussion is added in the revised manuscript.

Page 19: “The global segregation index over the entire casting volume ($GMI = \frac{\int_{Vol} \bar{c}_{mix}^{index} dV}{\int_{Vol} dV}$) was 3.29% for the case in which fragmentation was ignored, which is similar to the value of 3.32% for the case with fragmentation. Considering spurious grains has a very limited effect on the simple-shaped casting with cross-sectional changes, but it has a significant effect on the industrial-scale turbine blade casting with complex geometries and thin-wall structures. More discussion is provided in Section 5.4.”

Page 24: “Compared to our previous research [25], it was found that considering spurious grains has a significant impact on the prediction of channel segregation. In earlier two-phase simulations that ignored spurious grains, a segregation channel was predicted on the concave side of the blade casting, but this channel was not observed experimentally. The current three-phase model, which accounts for spurious grains, resolves this discrepancy between experimental results and simulation predictions. Another major difference lies in the shape and distribution of the segregation channels on the convex surface. In Figure 11(a), the fish-tail-shaped region (in yellow colour) at the bottom of the casting indicates the competition and merging of plume flows during solidification [2,11] before developing into a stable segregation channel. This phenomenon was not predicted by the two-phase model. The pronounced effect of considering the spurious grains on the prediction of channel segregation in blade castings may be attributed to the fact that, in thin-walled castings, even a minor influence of spurious grains on fluid flow can significantly impact the initiation and further development of freckles. Essentially, neglecting the formation of spurious grains only allows for the prediction of channel segregation. It is only by considering the formation, movement, and solidification of spurious grains that freckles can be truly predicted.”

(4) The relationship between eutectic and freckle is closely related, based on the existing simulation results. However, these relationships are embedded within the text and not explicitly expressed. Could the author provide a clear and concise explanation of this connection based on the available results?

Answer: Yes! It was added.

Page 16: “After the casting has fully solidified, the solute-enriched liquid in the segregation channels solidifies as eutectics, with spurious grains embedded within it, a phenomenon referred to as freckles.”

Reviewer #2:

1. Page 10: Why was the Heaviside step function chosen to describe the probability of fragmentation?

Answer: In this model, the Heaviside step function acts as a switch, transmitting a unit signal when a specified condition is met.

2. Figure 7: The results show that the number of fragments from flow-driven fragmentation is much greater than that from capillary-driven fragmentation. However, previous studies indicate that the primary source of dendrite fragmentation is capillary-driven (T. Cool, Acta Mater., 2017).

Answer: Thank you very much for your comment! Cool and Voorhees performed isothermal coarsening experiments on board the ISS (international space station). In their experiments, the flow is ignorable, and the mass transfer is governed by solute diffusion in the interdendritic liquid. Thus, the fragmentation is dominated by the capillary-driven mechanism. In the current study, the liquid flow in the segregation channels is significant. The mass transfer via the melt convection is several orders of magnitude faster than diffusion. Hence, the fragmentation rate via the flow-driven mechanism is much greater than that from capillary-driven mechanism.

3. Figure 9: Considering the formation of spurious grains seems to have no impact on the predicted distribution of freckles. Does this mean that spurious grains can be ignored in future freckle simulations?

Answer: Thank you very much for your comment! In castings with simple shapes (i.e. the square prism casting with cross-section change), considering spurious grains seems to have little impact on the predicted segregation channels. However, on a real turbine blade casting with complex shape and thin-wall structure, considering spurious grains has significant effect on the accuracy of the modeling results. New discussion has been made in the revised manuscript. We think it is important to consider the spurious grains in future freckle simulations, especially for real turbine blade castings.

Page 19: “The global segregation index over the entire casting volume ($GMI = \frac{\int_{Vol} \bar{c}_{mix}^{index} dV}{\int_{Vol} \bar{N} dV}$) was

3.29% for the case in which fragmentation was ignored, which is similar to the value of 3.32% for the case with fragmentation. Considering spurious grains has a very limited effect on the simple-shaped casting with cross-sectional changes, but it has a significant effect on the industrial-scale turbine blade casting with complex geometries and thin-wall structures. More discussion is provided in Section 5.4.”

Page 24: “Compared to our previous research [25], it was found that considering spurious grains has a significant impact on the prediction of channel segregation. In earlier two-phase simulations that ignored spurious grains, a segregation channel was predicted on the concave side of the blade casting, but this channel was not observed experimentally. The current three-phase model, which accounts for spurious

grains, resolves this discrepancy between experimental results and simulation predictions. Another major difference lies in the shape and distribution of the segregation channels on the convex surface. In Figure 11(a), the fish-tail-shaped region (in yellow colour) at the bottom of the casting indicates the competition and merging of plume flows during solidification [2,11] before developing into a stable segregation channel. This phenomenon was not predicted by the two-phase model. The pronounced effect of considering the spurious grains on the prediction of channel segregation in blade castings may be attributed to the fact that, in thin-walled castings, even a minor influence of spurious grains on fluid flow can significantly impact the initiation and further development of freckles. Essentially, neglecting the formation of spurious grains only allows for the prediction of channel segregation. It is only by considering the formation, movement, and solidification of spurious grains that freckles can be truly predicted.”

4. The calculation of a single case took 10 days on a high-performance cluster. With the advancement of computer technology, this calculation time may be reduced in the future. In addition to improvements in computational capacity, are there any possibilities to enhance the entire VA-model algorithm to accelerate the calculation?

Answer: Yes! We are working on optimizing the calculation algorithm. Some ideas like using adaptive time-step and dynamic adaption of the mesh are trying now to improve the efficiency of the solidification code.

Reviewer #3:

Minor remarks:

1. Line 108, in the text is "central rod" while it is not included in the model presented in figure 1b. Also, there is a hole in the chill plate and mold. Why was the central rod removed and the hole in the chill plate included in the simulation model?

Answer: Thank you very much for your comment!

There was a mistake in the original manuscript. We have confirmed with our colleagues who performed the directional solidification experiments. The "central rod" was not assembled for this set up of experiments. This mistake has been corrected.

Page 4: "A cluster of superalloy castings was assembled around the **axis of the furnace...**"

In the experiment, the castings and molds were placed on a transition chill plate (orange), on which there is a hole in the center, as shown in the schematic figure below. The transition chill plate was connected to a round plate (blue). This round plate was not considered in our simulation. According to our previous experience, such simplification almost has no effect on the modeling results.

2. Line 113, the temperature of the melt poured into the mold cavity should be given.

Answer: The temperature of the melt is the same as the furnace temperature (1750 K). This has been added in the revised manuscript.

Page 4: "The moulds were preheated to 1750 K in the furnace, and the hot melt (1750 K) was then poured into the moulds."

3. It would be good to show the shadow side on the drawing of the furnace (Fig 1).

Answer: Thank you for your suggestion. The shadow side of the furnace is marked in Figure 1(b).

4. Line 148. “(d1)-(d3) Microstructure on tranverse sections”. Microstructures (d1) and (d2) are not included in Fig. 2. Please correct this.

Answer: Thank you very much for your comment! This has been corrected!

5. The two areas marked with red dashed lines in Fig 2c are not described in the text (segregation channels?).

Answer: Thank you very much for your comment! Actually, the two areas marked with red dashed lines in Fig 2(c) was mentioned in section 5.4: “The same phenomenon can be seen in **Figure 2(c)**. The segregation channels below the freckles are delineated by red dashed lines.” But, we agree with the reviewer! It is necessary to describe this segregation-to-freckle-transition in section 2. New description has been added in the text.

Page 5: “The dark-coloured freckles originated from the segregation channels, which are marked by the red dashed lines in **Figure 2(e)**. The Electron Backscatter Diffraction (EBSD) detection was performed on the region marked in blue in **Figure 2(e)**. The detected results are shown in **Figure 2(f)**. Numerous small spurious grains are dispersed within the freckles, whereas the segregation channels are free of spurious grains but exhibit an accumulation of eutectics.”

6. According to the description in the text, 12 castings were manufactured. What was the basis for selecting this one representative casting shown in Fig. 2?

Answer: A cluster of superalloy castings (12 in total) with different shapes (cylinder, square prism, zig-zag shape and flower shape, etc) were casted in one batch. Freckles were formed on all samples, and the phenomenon that a freckle developing from a segregation channel was observed on all samples.

There are two main reasons to select this casting for this research: first, it is possible to study the effect of the cross-sectional change on the development of the casting defects; second, compared to samples with a curved surface, it is easier to polish and measure the structure on a flat casting surface.

7. Line 202, “uc” not added in Fig. 3a.

Answer: “uc” has been added in the revised manuscript on Page 3 in Fig. 3(a).

8. Better modify Fig. 3a to show directional solidification in a vertical configuration and columnar dendrites, which are more consistent with experiment and simulations for the Bridgman process.

Answer: Thank you very much for your insightful comment. Figure 3 has been revised on Page 3.

9. Line 311, from which area is the solute-enriched liquid sucked into the channels, the interdendritic area?

Answer: Yes! The solute-enriched liquid was sucked into the channels from surrounding interdendritic area. The manuscript has been revised to make it more understandable.

Page 14: “In return, the liquid flow further updated the c_i and ρ profiles by sucking the solute-enriched liquid into the channels from their surrounding interdendritic area. The solute-enriched liquid was then transported upward in the channels and plumes.”

Major remarks:

1. Most of the simulations were carried out for the lower segments of the casting, whereas the microstructure of the freckles and channels is only shown for the top segment. It would also be good to show the microstructure of the freckles on the surface of the lower segment (lower casting area) in order to compare experimental and simulated results. I think this would be more interesting for the reader. The morphology of the freckles in the lower segment of the casting is very poorly visible in Fig. 1a.

Answer: We completely agree with your comments! The microstructure of the freckles on the surfaces of the two lower segments is shown in the revised manuscript on Page 6.

2. According to Fig.4c, for a time of 4000 s, the temperature at points A and B reaches a value of about 1650 K, whereas the temperature value in Fig.4b for these points is much less than 1600 K (same time of 4000 s). These values are not consistent in the two figures for the same times. Please check this.

Answer: Thank you very much for figuring out this mistake! It has been corrected in the revised paper on Page 13.

3. In my opinion, it should be emphasized in the text that the solidification times given for the simulations shown in the Figs. 5 and 6 do not take into account the solidification stage of the starter and selector. This can be confusing for the reader, especially since simulation results (Fig 4a) obtained in ProCAST software are also shown in the paper.

Answer: We have checked. The simulation results of both ProCAST and the multiphase solidification model refer to the same time. In other words, the solidification times given for the simulations shown in Figs. 5 and 6 did take into account the solidification stage of the starter and selector. The simulation began with the withdrawal of the sample and cooling system, and the stabilization time following pouring was not counted in the current simulation. This has been mentioned in section 3.3.

Page 11: “The stabilization time following pouring was not counted in the current simulation.”

4. Line 396 “The upper part of the casting was more susceptible to freckles than the lower part “. It would be interesting for readers to explain in more detail in the paper why the propensity to freckle formation is higher in the upper part of the casting than in the lower part. From my observations it seems that the curvature of the liquidus isotherm is usually highest at the top of the casting. Perhaps this is related to the freckle formation.

Answer: Thank you very much for the comment! A new explanation has been added in the revised manuscript in section 4.4.

Page 18: “As the casting is withdrawn, the downward movement of the water-cooled copper plate reduces the isolation between the furnace's hot and cold zones. This decreases the temperature gradient at the solidification front, creating more favourable conditions for the formation and development of freckles. Consequently, the upper part of the casting was more susceptible to freckles than the lower part.”